# Comparison of Brazilian Social Interest Housing Projects Considering Sustainability

**DOI:** 10.3390/ijerph19106213

**Published:** 2022-05-20

**Authors:** Paulo Cezar Vitorio Junior, Víctor Yepes, Moacir Kripka

**Affiliations:** 1Graduate Program in Civil Engineering, Federal University of Technology—Paraná, Via do Conhecimento, Km 1, Pato Branco 85503-390, PR, Brazil; paulocjunior@utfpr.edu.br; 2Institute of Concrete Science and Technology (ICITECH), Universitat Politècnica de València, 46022 Valencia, Spain; vyepesp@cst.upv.es; 3Graduate Program in Civil and Environmental Engineering, University of Passo Fundo, Km 292, BR 285, Passo Fundo 99052-900, RS, Brazil

**Keywords:** social interest housing, life cycle thinking, analytic hierarchy process, sustainability

## Abstract

Considering the importance of the development of new housing projects, the purpose of this research is to provide a model oriented to the identification of the most sustainable alternative in single-family housing projects of social interest from the perspective of life cycle thinking (LCT) and the analytical hierarchical process (AHP). A ceramic masonry project and a concrete masonry project were evaluated. In the environmental dimension, the results showed that the ceramic masonry project had more significant environmental impacts and greater damage to human health and the availability of resources and ecosystems. In the social dimension, it was found that there are discrepancies between the salaries in the construction supply chain and that the concrete masonry project had better social characteristics than the ceramic masonry project. The economic dimension revealed that the concrete masonry project was more attractive. Relating the environmental, social, and economic dimensions’ results, through the combination of LCT and AHP, it was found that the concrete masonry project presented a combination of more sustainable characteristics than the ceramic masonry project in the majority of the results. Among the implications of the study carried out here is the advancement of sustainability applied to the construction sector.

## 1. Introduction

The construction industry is related to significant impacts [1,2,3], such as the use of energy and emissions [4,5,6,7,8], natural resources consumption [9,10,11,12,13,14,15], and waste generation [16]. In addition, construction has significant social impacts on human populations [17,18,19,20,21].

Despite the impacts, the construction sector is one of the main supports for economic development and social well-being, especially in developing countries [22,23,24]. In 2019, the Brazilian construction sector totaled USD 72 billion in activities and reached USD 11.5 billion in salaries and other compensation [25]. On average, construction accounted for 5.3% of the Brazilian GDP between 2011 and 2019, highlighting the importance of the construction sector to the Brazilian economy [26].

Housing is the product with greater participation in the Brazilian construction sector; it is responsible for about 26% of buildings. With economic development, housing consumes about 34% of Brazilians’ income [27]. The latest results estimated a Brazilian deficit of 5.876 million households. About 14.257 million households had inadequate at least one type of infrastructure service, corresponding to 22.8% of the permanent private urban households in the country [28,29].

The demand for housing generates and sustains new market opportunities for the construction sector, particularly for social housing projects [30]. The construction of new houses requires replacing traditional techniques with more sustainable construction systems [21,31,32,33] and materials [34,35]. Over the years, economic issues are almost exclusively used to make project decisions. In many cases, the environmental dimension is the best sustainable solution when a sustainable construction must include the triple bottom line assessment (environmental, economic, and social dimensions) [36,37,38].

Given the presented scenario, there is an opportunity for scientific advancement that adds knowledge to sustainable housing construction. It is possible to verify the importance of developing tools that support designers’ decision making to generate more sustainable houses.

This paper proposes a methodology to evaluate sustainability in Brazilian social interest housing projects using life cycle thinking and the analytic hierarchy process. The methodology was applied to two house projects: a ceramic masonry project and a concrete masonry project. Based on the results obtained from environmental, economic, and social assessments, the analytic hierarchy process allows the comparison of criteria to identify the best global alternative. Based on the obtained results, it is also expected to point out sustainable improvements in housing projects, considering that sustainable projects can make the habitations more appropriate, bringing security, comfort, privacy, and positive impacts to their inhabitants’ physical and mental health [39].

Life cycle thinking (LCT) approaches provide a framework for integrating environmental policies and development strategies [40,41,42]. It is a valuable tool in conceptual design and decision making [43]. The methodology assesses the impacts of a product throughout its life cycle, overcoming the conventional approach centered on manufacturing processes and the production site. The principal goals of an LCT approach are to reduce resource use and the emissions of products and to improve socio-economic performance throughout their life cycles [44]. LCT approaches can be used in all sectors, and offer the possibility to examine a series of categories and key impact indicators, assessing environmental, economic, and social impacts and their final effects on the dimensions of sustainability [45]. Among the possible approaches to LCT are life cycle assessment (LCA), social lifecycle assessment (SLCA), and Life Cycle Cost (LCC).

A lifecycle assessment (LCA) estimates the possible impacts on the environment over the life cycle of a product [46,47,48]; helps in the identification of components, processes, and systems; compares options to minimize environmental impacts; and provides advice on strategic and long-term planning concerning trends in materials and products design [49,50].

The International Organization for Standardization (ISO) formalized the LCA in its 14,040 series [49,50]. The LCA is organized into four separate parts: objective and scope, inventory analysis, impact assessment, and interpretation. All phases of the LCA are interrelated, which indicates that the process of conducting an LCA is dynamic and iterative.

Asocial life cycle assessment (SLCA) identifies, communicates, and reports potential social impacts and social conditions [51]. It evaluates the performance of an organization, product, or system on these different stakeholders. An SLCA verifies the possible impacts that can influence firm performance and the impacts on social capital over a life cycle, and it is usually generic and data-driven at a particular site [15]. It emphasizes human welfare and the facilities adopted to raise life quality.

A life cycle cost (LCC) assessment evaluates and summarizes the project’s total cost. It considers all costs of acquiring, owning, and disposing of a product to support a financial decision. The purpose of an LCC is to evaluate the total costs of the project’s alternatives and select the design that ensures that the installation offers the lowest overall cost consistent with its quality and function [52]. What differentiates the application of the LCC from other performance studies is that other costs during the life cycle phases are considered in addition to investment costs. The sum of all costs allows you to select the most financially advantageous alternative. An LCC comprises the following steps: (i) definition of the object of study; (ii) estimating the life cycle of the study elements; (iii) elaboration of the life cycle inventory of each of the components; and (iv) financial update and composition of total costs

When several alternatives and multiple criteria need to be considered, using multicriteria decision-making methods (MCDM) is essential. They may be applied to different fields [53,54,55], allowing the decision maker to assess intricate problems involving complex problems involving multiple and conflicting criteria based on the subjective opinions of a group of experts or interested parties [56].

Among the MCDM methods, analytic hierarchy process (AHP) is the most widespread. AHP is an approach for assessing objective and subjective functions in multicriteria decision making and helping users achieve an appropriate decision. An essential feature of AHP is achieving agreement in the decision-making process in a group [57]. The main reasons for the widespread use of AHP are its capacity to assess objective and subjective measurements, its simplicity and versatility, its decomposition of a complex problem into several criteria hierarchies, and its ability to check the inconsistencies in a decision maker’s opinions [58]. 

## 2. Methodology

The development of the sustainability assessment was initiated with the research and selection of representative projects of social interest of different construction types. Two projects were selected: a ceramic masonry project and a concrete masonry project. These projects were obtained from public agencies, and their typologies were adopted because they corresponded to the vast majority of social interest housing in Brazil. A life cycle assessment (LCA), social life cycle assessment (SLCA), and life cycle cost (LCC) assessment were performed to evaluate the projects regarding LCT. Following these, the multicriteria decision-making method AHP was applied, aiming to identify the best project according to several different perspectives. The following items detail the adopted methodology and the results obtained from its application.

### 2.1. Selection of Projects

Figure 1 and Figure 2 present, respectively, the ceramic masonry project and the concrete masonry project adopted in the present study. 

The projects comprise ceramic masonry (with a constructed area of 49.09 m^2^) and concrete masonry (with a constructed area of 44.78 m^2^), with the system specified in Table 1.

### 2.2. Sustainability Assessment

#### 2.2.1. Conducting the Life Cycle Assessment

(a)Definition of the objective and scope

An LCA was applied to both projects to identify environmental impacts and compare environmental performance throughout their life cycles. The houses evaluated were to protect and provide comfort and safety to a family of four inhabitants. The functional unit employed was 1.0 m^2^ of housing built for 50 years.

The product system presents the elementary processes that make up the housing life cycle. The system boundary defines life cycle stages and which processes belong to the analysed system. In this research, an evaluation is carried out from the cradle to the gate. As the system’s frontier, the pre-operational phase is considered, where the processes of extraction of raw materials, manufacture of construction materials, and construction of housing are located. The construction comprises the infrastructure (foundation); superstructure; internal and external sealing (walls); roof; and coatings, linings, and paint (finishes). The pre-operational phase was chosen once in this phase, and the designer can make changes to increase sustainability. Figure 3 shows the product system and boundary employed for the LCA of the projects.

(b)Life Cycle Inventory Analysis

This phase involves compiling and quantifying inputs and outputs related to the materials. The projects were modelled in a software system created for the integrated environmental assessment of products (SimaPro) and followed the structure of the codes from ISO 14040:2006 and ISO 14044:2006. The data were extracted from the international database ecoinvent v3.3, and the processes used were of the market type. Table 2 presents the materials listed in the inventory and their respective SimaPro processes.

(c)Life Cycle Impact Assessment

For the LCIA of the housing projects, the ReCiPe 2016 characterization method was adopted in its hierarchical perspective and endpoint approaches.

(d)Interpretation

At this stage, important aspects of the LCA were identified from the results of the LCI and the LCIA, and the conclusions and limitations of the study were listed.

#### 2.2.2. Conducting the Social Life Cycle Assessment

This paper used the Fair Wage Potential [59] and Weighted Fair Wage Potential [60] in the social assessment.

(a)Objective and scope

The data were obtained from a national survey developed by a Brazilian government institute (IBGE). The social data (2002 to 2015) considered the timber and steel industries and non-metallic and chemical inputs.

(b)Social Inventory of the Life Cycle

Materials were classified based on their sector in the construction supply chain through the material inventory. Table 3 displays the selected materials.

(c)Fair wage potential

Fair Wage Potential (*FWP*) was proposed as a model to convert the materials’ data on salary to workers and the product’s life cycle into a category indicator. *FWP* considers the wage paid at each process as the minimum living wage. It considers a relation between paid wage and the working time in each sector, as stated in Equations (1) and (2).
(1)$$FWP_{n}=\frac{RW_{n}}{RWT_{n}}\times \left(CF_{FW,n}\right)$$
(2)$$CF_{FW,n}=\frac{1}{MLW_{n}}\times CWT_{n}\times \left(1-IE{F_{n}}^{2}\right)$$
where:

$FWP_{n}$: Fair wage potential (expressed in $FW_{eq})$ for process n in the life cycle of a product in a defined location/sector;$RW_{n}$: Real wages (R$ per month) paid to workers employed in process n over a year;$RWT_{n}$: Real working time (hours per week) of workers employed in process n (including vacation and unpaid overtime);$CF_{FW,n}$: Fair wage related characterization factor (month per R$) for process n at the defined location/sector.$MLW_{n}$: Minimum living wage (R$ per month) that must be paid to allow an appropriate standard of living for an individual or family in the respective place/sector;$CWT_{n}$: Contracted working time (hours per week) to perform the process n; and$IEF_{n}$: Factor of inequality (in percentage) at the place/sector that performed the process n.

Real wages $\left(RW_{n}\right)$ corresponds to the monthly income received by the worker. For the minimum living wage $\left(MLW_{n}\right)$ the Brazilian values can be viewed in Table 4.

The study considered for ${CWT}_{n}$ the duration of the weekly workday as 44 h. For $RWT_{n}$, the number of hours worked weekly was obtained from the Brazilian Government Institute (IBGE). As the inequality factor $\left(IEF_{n}\right)$, the study considered the Gini index values indicated in Table 4.

(d)Weighted fair wage potential

The weighted fair wage potential (*WFWP*) is a method proposed to associate the material inventory to the social data sector. The total mass of a given product can be calculated considering the supply chain sectors, allowing the obtention of the sector equivalent mass (SEM). Equation (3) indicates the $WFWP_{n}$ calculation:(3)$$WFWP_{n}=\frac{1}{\sum SEM_{n}}\;\overset{NoS}{\underset{i=1}{\sum }}{\left(SEM_{n}\right)}_{i}\times {\left(FWP_{n}\right)}_{i}$$
where:

i: Sector;NoS: Number of sectors;$WFWP_{n}$: *WFWP* of process n of a product in a location/sector;$FWP_{n}$: *FWP* of process n of a product in a location/sector; and$SEM_{n}$: Sector equivalent mass of material for process n of a product in a location/sector.

*SEM* was calculated from the materials’ inventory (Table 5), representing the amount of material corresponding to the sector (1.0 m^2^ of the construction).

The *WFWP* for ceramic and concrete masonry projects was calculated by relating *FWP* and *SEM*. The results relate to the sector’s social situation and the projects’ social assessment.

#### 2.2.3. Conducting the Life Cycle Cost

The project cost survey was built based on the materials inventory, which considered the acquisition costs (AC) and installation costs (IC). The economic LCI was built from the architectural and service projects of the social interest housing projects. The primary materials present in the chosen life cycle stages were included. The cost survey employed was based on compositions established by the Brazilian Federal Government.

#### 2.2.4. Application of AHP 

To develop a comprehensive comparison of ceramic and concrete masonry projects, it is essential to adopt a multicriteria decision-making method such as the AHP once it allows the consideration of conflicting criteria. For example, a good solution from a social perspective can be the most expensive, and vice versa. 

Despite the availability of software devoted to performing the AHP, the formulation of the method was implemented by the authors as an MS Excel sheet.

Although AHP can deal with subjective criteria, only quantitative criteria were considered in this work once the LCA, SLCA, and LCC could be quantified as described in previous items. In the original method, a pairwise comparison of alternatives regarding each criterion had to be performed according to the Saaty fundamental scale. The impact value of alternative j about alternative i is the numerical representation of the attribution given by the decision maker to each comparison of alternatives. In the present work, after the quantification of each alternative (e.g., the cost of each project), the corresponding values are normalized, representing the relative importance of the alternatives regarding the considered criteria. Similarly, each criterion’s weights can be determined by using a pairwise comparison and the fundamental scale of Saaty. The weights were directly taken from the ReCiPe 2016 characterization method in this work.

## 3. Results and Discussion

### 3.1. Life Cycle Assessment

#### 3.1.1. Life Cycle Inventory

Table 6 shows the LCI for the ceramic and the concrete masonry projects, per square meter.

#### 3.1.2. Life Cycle Impact Assessment and Interpretation

The damage analysis verified that the ceramic masonry project presented a more significant generation of damages to human health, ecosystems, and resources. This can be seen in Table 7 and Figure 4.

### 3.2. Social Life Cycle Assessment

#### 3.2.1. Fair Wage Potential

Figure 5 shows the fair wage potential calculated from 2002 to 2015.

Figure 5 indicates that the considered sectors presented a fair wage potential higher than the unit, indicating that the real wages exceeded the minimum wage. The steel industry paid the highest wages, followed by chemical and non-metallic inputs. All sectors worked less than forty-four hours weekly (chemical and non-metallic inputs worked more hours, followed by the steel industry). 

#### 3.2.2. Weighted Fair Wage Potential

The FWP indicates the relative situation of each sector, but it is not possible to apply these results directly to ceramic and concrete projects. On the other hand, the weighted fair wage potential condenses the materials used and the social situation of the workers in a single value. The FWP analyses the presence of each sector regarding each project, as seen in Table 5.

Figure 6 outlines the WFWP values for the considered projects. It reveals that the concrete masonry project had a higher weighted fair wage potential than the ceramic masonry project. The concrete project had a significant percentage of materials in sectors with the highest FWP (steel, chemical, and non-metallic). 

#### 3.2.3. Life Cycle Cost

For project-related costs, the economic LCI was built based on the inventory of materials. The costs for the functional unit were obtained from the Brazilian market. The acquisition and installation costs were taken into account, and costs with materials, labour, and equipment were included.

Based on previous considerations, it was observed that the total cost per square meter of a ceramic masonry project (USD 230.52) was higher than the cost of a concrete masonry project (USD 221.53). This made the concrete masonry project economically more attractive.

When analysing the costs per construction phase, it appeared that the ceramic masonry project was economically more advantageous in the infrastructure (ceramic masonry USD 23.58 × concrete masonry USD 37.02) and walls (ceramic masonry USD 47.99 × concrete masonry USD 58.81). The concrete masonry project was more advantageous in its superstructure (ceramic masonry USD 37.80 × concrete masonry USD 19.68), roof (ceramic masonry USD 5.56 × concrete masonry USD 36.79), and finishes (ceramic masonry USD 79.60 × concrete masonry USD 69.23).

### 3.3. Sustainability Assessment

This study considers the environmental, social, and economic dimensions, presenting the model’s results for identifying the most sustainable alternative in single-family housing projects of social interest from the perspective of LCT and AHP. For each of the analyses carried out, the proposed model determines which of the considered projects presents more sustainable characteristics, helping the designer’s decision.

The complex data obtained by the LCA, SLCA, and LCC are compiled and related using the AHP method in the proposed model. With this methodology, the designer can evaluate the projects considering the hierarchical, egalitarian, and individualistic cultural perspectives and the concept of ideal sustainability with equal weights. Table 8 shows the used weights in the sustainability assessment.

The environmental dimension variables contain data from the environmental analysis using the ReCiPe 2016 characterization method in its hierarchical, egalitarian, and individualist perspectives and endpoint approaches. It is important to stress that other weights to each criterion can be determined through AHP, according to the designer’s experience or interviews with experts. The decision maker can make this through a pairwise comparison of criteria using the fundamental scale of Saaty. In this work, the well-established ReCiPe 2016 method was considered instead.

The *WFWP* gives the social dimension for the year 2015. The total cost per square meter gives the economic dimension.

Table 9 presents the values of the environmental, social, and economic criteria used to determine the sustainability of the projects from a hierarchical perspective, obtained according to previous items. Table 10 presents the model’s results for sustainability from a hierarchical perspective, considering the relative importance of criteria regarding each alternative.

Table 10 shows that from a hierarchical perspective, the concrete masonry project (56.19%) presents a more sustainable combination of environmental, social, and economic characteristics than the ceramic masonry project (43.81%).

Table 11 presents the values of the environmental, social, and economic criteria obtained from the egalitarian perspective.

Table 12 shows that from an egalitarian perspective, concrete masonry (52.55%) presents a more sustainable combination of environmental, social, and economic characteristics than ceramic masonry (47.45%).

Table 13 presents the values used to determine sustainability from an individualistic perspective, and Table 14 shows that concrete projects are more sustainable than ceramic projects (55.43% and 44.57%, respectively).

Finally, Table 15 presents the values of the environmental, social, and economic criteria used to determine the sustainability of the projects from the concept of ideal sustainability that considers equal weights.

Table 16 shows that from the ideal sustainability concept that considers equal weights, the concrete masonry project (55.04%) presents a more sustainable combination of environmental, social, and economic characteristics than the ceramic masonry project (44.96%).

Table 17 shows the summary of the sustainability assessment. The results show that the concrete masonry project represents a combination of environmental, social, and economic characteristics that are more sustainable than the ceramic masonry project in all performed analyses.

Based on the obtained results, the designer can choose the concrete masonry project to implement or make adjustments to the ceramic masonry project. The designer has a direct influence on the design and dimensioning phases. Then, according to the level of sustainability obtained, the project can be modified by changing the quantities of material, the types of material, the geometry of the project, and even changing the constructive typology.

As previously stressed, the pre-operational phase was chosen for this study once in this phase, and the designer can make changes to increase sustainability. Due to this, the production phase constitutes a promising area of research and contribution to mitigating the impacts. On the other hand, it could be desirable to evaluate materials and systems from a full life cycle view once a good solution from a production perspective could significantly impact the use stage. In addition, it is also important to stress that the inclusion of other critical criteria, such as the energy efficiency of structural materials, can affect the obtained results [56,62,63,64].

In order to enable the proposed study, some limitations and definitions were necessary. Among these, were the definition of the number of criteria and the way they were quantified. For example, the consideration weighted fair wage potential to social life cycle assessment was adopted, as in previous studies developed by authors, as proposed in [58]. The results can present significant variations if other stakeholders are considered (such as the local community) [65].

The weighting of the sustainability dimensions is generally based on the weights determined through questionnaires and interviews with professionals and specialists. The samples of consulted professionals must be composed of many elements to reduce the subjectivity, which is a limitation in the data collection process due to the high costs and the amount of time required. The cultural perspectives and equal weights used here to weight the sustainability dimensions reduce the subjectivity of the analysis, as they do not depend on the view or opinion of other professionals. They are widely used in software involving LCT, such as ReCiPe 2016 and the Eco-Indicator 99 characterization methods, which are globally used and accepted by the scientific community and industry, proving that the use of cultural perspectives is advantageous. 

## 4. Conclusions

This study presents a project evaluation model focused on sustainability that was validated by analyzing two single-family housing projects of social interest of different construction types (ceramic masonry and concrete masonry). The proposed tool aggregated data on the dimensions of sustainability and enabled the generation of decision-making scenarios involving environmental, social, and economic issues.

Designers and decision makers in the civil construction sector in public and private spheres can use this model to verify potential points of improvement for a projects’ life cycle. The use of life cycle thinking as criteria and sub-criteria of the hierarchical analytical process makes it possible to identify the most significant modifications that can be carried out in the projects, and they should be the focus of analysis by the designer in the design and dimensioning processes, as follows:-In the environmental dimension: the results obtained through the ReCiPe 2016 method showed that the concrete masonry project generates less impact and minor environmental damage when compared to the ceramic masonry project.-In the social dimension: the potential fair wage showed differences between the wages in the construction sectors. The weighted fair wage potential showed that the concrete masonry project had better social characteristics than the ceramic masonry project during the analysed period.-In the economic dimension: the results pointed out that from the point of view of the total cost per square meter, concrete masonry is, economically, more attractive. The ceramic masonry project is economically more advantageous in the infrastructure and wall phases during the construction phase. The concrete masonry project is more economically advantageous in the superstructure, roofing, and finishes phases.-Relating the results of the environmental, social, and economic dimensions through the combination of LCT and AHP: it was found that the concrete masonry project presents a combination of more sustainable characteristics than the ceramic masonry project in the majority of the results.

Among the implications of the study carried out here is the advancement of sustainability applied to social interest housing projects. The model and results presented make it possible to complement the existing quality assessment programs in construction and contribute to generating a new evaluation and certification methodology for projects involving environmental, social, and economic issues. The literature review pointed out that most studies consider the environmental dimension as absolute sustainability. For these cases, this study can contribute as a basis for incorporating other dimensions of sustainability by including weighted fair wage potential as a social dimension and the cost of the life cycle as an economic dimension, which will generate a complete sustainability result. The environmental dimension variables contained data from the environmental analysis using hierarchical, egalitarian, and individualist perspectives and endpoint approaches. Although other weights to each criterion can be adopted, this work considered the well-established ReCiPe 2016 characterization method.

In addition to verifying projects considering environmental, social, and economic issues, it is essential to highlight the link of this paper to the 2030 Agenda of the United Nations. In the environmental dimension, the verification of projects, when they generate impacts and environmental damage using the ReCiPe 2016 method, generates results that seek to solve the objective “Climate Action” (SDG 13), where measures are sought to combat climate change and its impacts.

In the social dimension, making wages more just and guaranteeing the workers’ rights within sectors that significantly influence projects can meet the objective “No Poverty” (SDG 1), which deals with ending poverty in all its activities. The fair wage potential calculated considering a worker’s wage assists in meeting the objective “Reduced Inequalities” (SDG 10), which is intended to reduce inequality within and between countries. For the objective “Decent Work and Economic Growth” (SDG 8), the results generated by the social dimension through the workers’ wages, as well as the results of the economic dimension where the most advantageous costs are presented, have a relation with the promotion of inclusive and sustainable economic growth, full and productive employment, and decent work for all.

By combining LCT with the AHP, the paper generates a new standardized and objective methodology for evaluating and designing projects. The results generated help to meet the objectives “Industry, Innovation and Infrastructure” (SDG 9), which deals with the construction of resilient infrastructures, promoting inclusive and sustainable industrialization, and fostering innovation; “Sustainable Cities and Communities” (SDG 11), which seeks to make cities and human settlements inclusive, safe, resilient and sustainable; and, “Responsible Consumption and Production” (SDG 12), which deals with ensuring sustainable production and consumption patterns. Although only two projects were considered in the present study, they represent most Brazilian social interest housing. In this sense, the same methodology can be easily expanded to consider other topologies and constructive aspects. 

Although the projects and data studied here are Brazilian, issues involving sustainability in housing projects of social interest are present in other developing countries due to demographic growth and urbanization. To verify international projects based on the proposed model, it should be checked for local environmental data. Social data related to local workers should be collected in the social dimension. In the economic dimension, cost surveys must be carried out according to the location to be studied. When combining LCT with the AHP, the analysis perspective should be selected to consider and apply the proposed methodology.

In this study, some aspects, such as the materials’ energy efficiency, were not included in the comparisons. On the other hand, due to the pairwise strategy adopted by the multicriteria decision-making method, this and other criteria (including those with more subjectivity) can be easily considered through the proposed methodology. 

## Figures and Tables

**Figure 1 ijerph-19-06213-f001:**
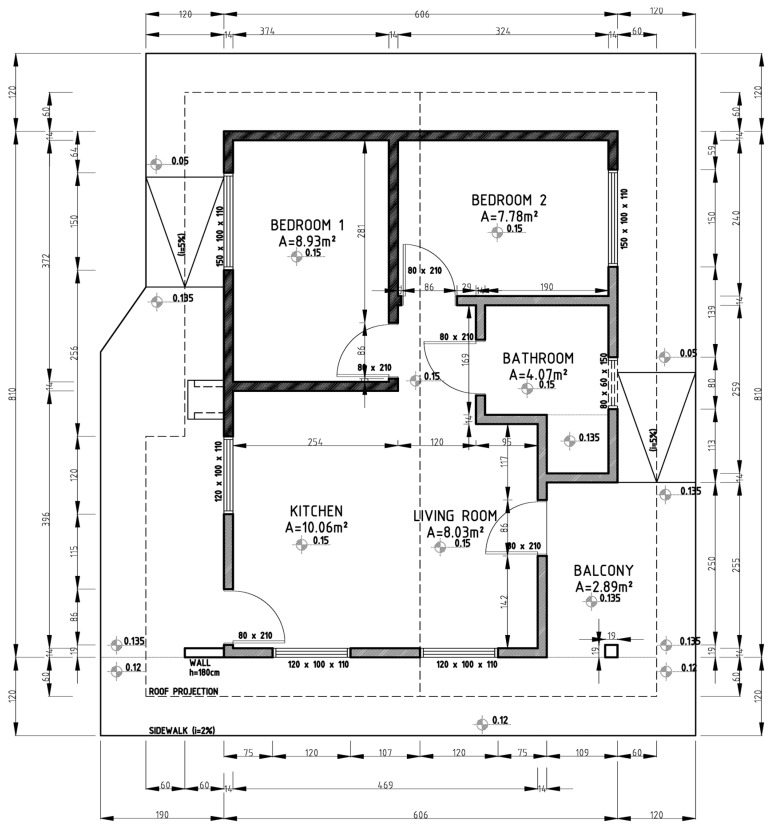
Brazilian social interest housing: ceramic masonry project (dimensions in cm).

**Figure 2 ijerph-19-06213-f002:**
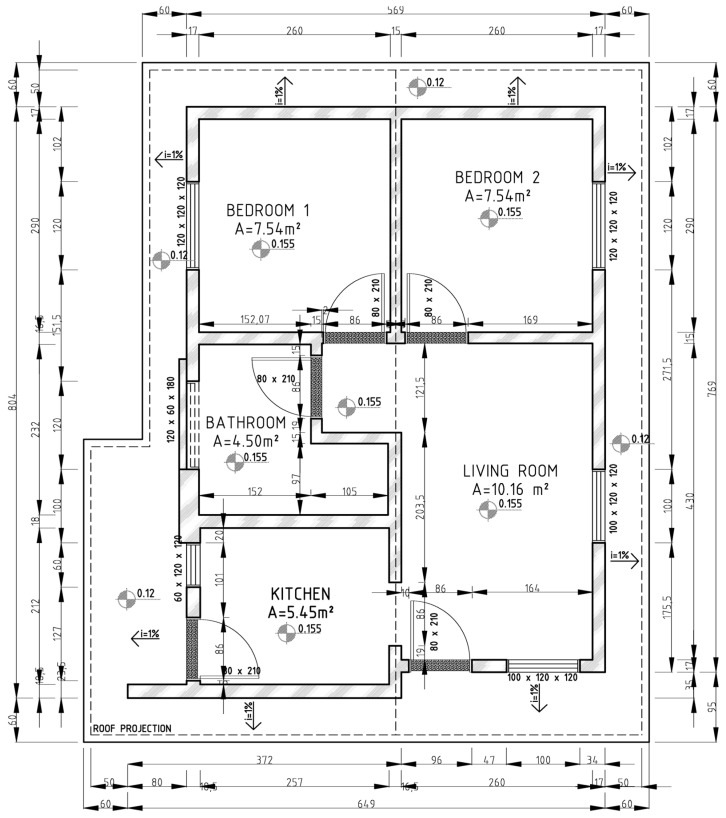
Brazilian social interest housing: concrete masonry project (dimensions in cm).

**Figure 3 ijerph-19-06213-f003:**
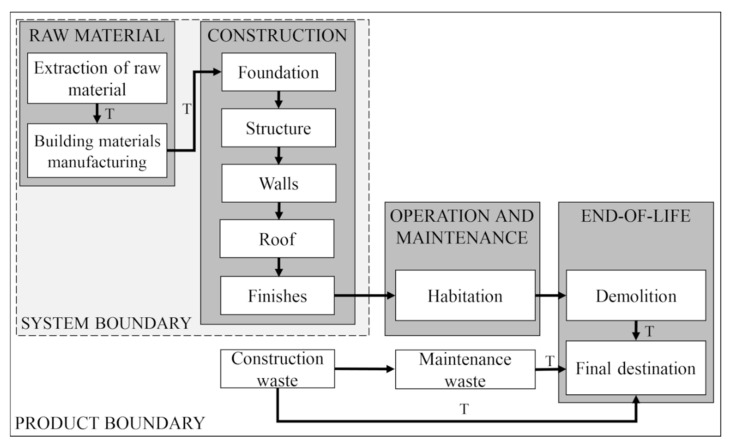
Product system and study system boundary.

**Figure 4 ijerph-19-06213-f004:**
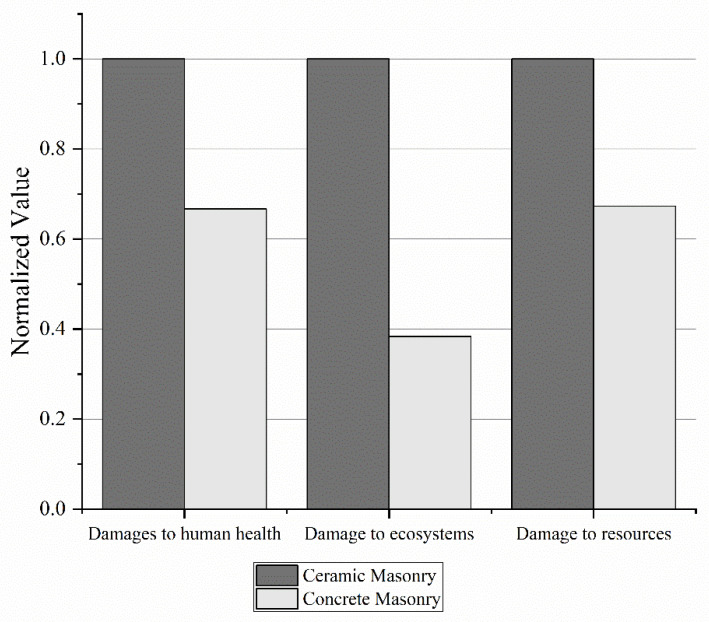
Normalized damage from ceramic masonry and concrete masonry.

**Figure 5 ijerph-19-06213-f005:**
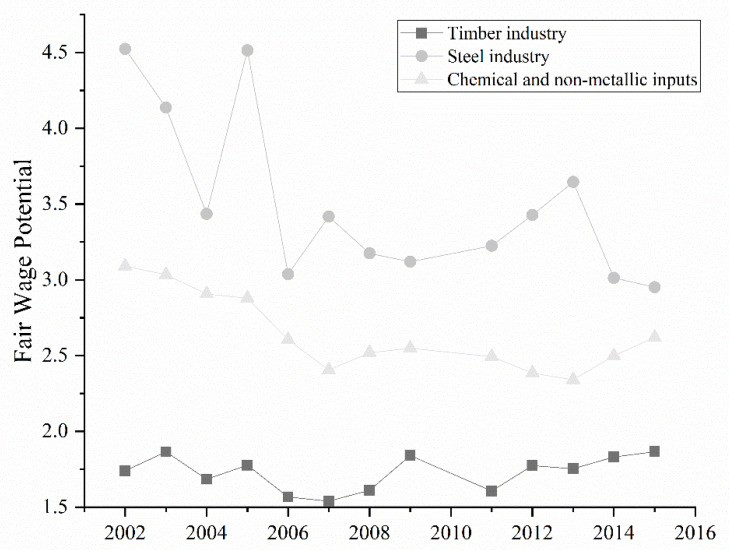
Fair wage potential for the considered industries.

**Figure 6 ijerph-19-06213-f006:**
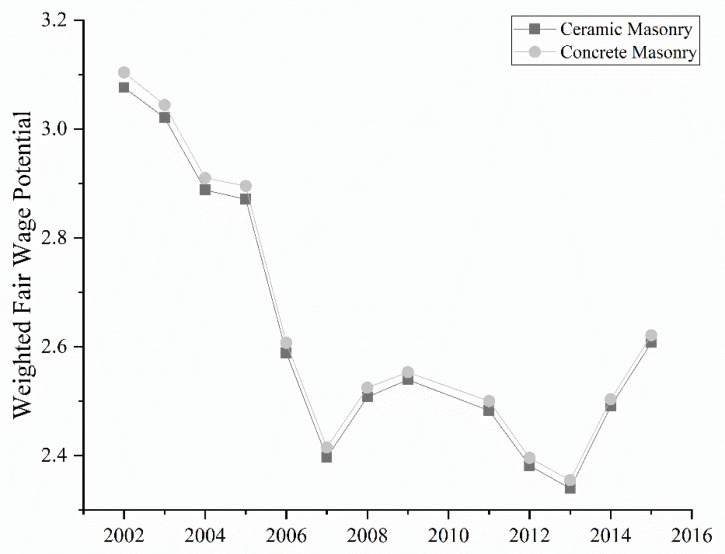
Weighted fair wage potential values for the ceramic masonry and concrete masonry projects.

**Table 1 ijerph-19-06213-t001:** Overall system of Brazilian Social Interest Housing Projects.

	Ceramic Masonry Project	Concrete Masonry Project
**Foundation**	Concrete drill piles; beams in ceramic masonry	Ceramic masonry
**Structure**	Ceramic masonry roof beams; ceramic masonry pillars and precast slabs	Precast ceramic masonry slabs; concrete blocks masonry (14 cm × 19 cm × 39 cm); cavities grout-filled; aluminium and wood frames
**Walls**	Clay blocks (9 cm × 14 cm × 19 cm) laid with lime mortar; aluminium and wood frame	
**Roof structure**	Wood	Steel
**Finishes**	Internal: cement and lime mortar; PVA mass; latex paint; ceramic skirting boards External: cement and lime mortar; PVC lining; acrylic paint	Internal: cement mortar and lime mortar; plaster cast; ceramic coating; latex paint, ceramic baseboards External: cement mortar and lime mortar; ceramic tile and acrylic paint

**Table 2 ijerph-19-06213-t002:** Materials and SimaPro processes.

Materials	SimaPro Process
Steel (kg/m^2^)	Reinforcing steel {GLO}|market for|Cut-off, U
Concrete (m^3^/m^2^)	Concrete, 20 MPa {GLO}|market for|Cut-off, U
Wood (m^3^/m^2^)	Sawn wood, Parana pine from sustainable forest management, kiln dried {GLO}|market for|Cut-off, U
Clay brick–massive (kg/m^2^)	Clay brick {GLO}|market for|Cut-off, U
Lime mortar (kg/m^2^)	Lime mortar {GLO}|market for|Cut-off, U
Cement mortar (kg/m^2^)	Cement mortar {RoW}|market for cement mortar|Cut-off, U
Clay brick–hollow (kg/m^2^)	Clay brick {GLO}|market for|Cut-off, U
Glass (kg/m^2^)	Flat glass, uncoated {GLO}|market for|Cut-off, U
Paint (kg/m^2^)	Alkyd paint, white, without solvent, in 60% solution state {GLO}|market for|Cut-off, U
Roof tile (kg/m^2^)	Roof tile {GLO}|market for|Cut-off, U
Ceramic tile (kg/m^2^)	Ceramic tile {GLO}|market for|Cut-off, U
Light mortar (kg/m^2^)	Light mortar {GLO}|market for|Cut-off, U
Concrete block (kg/m^2^)	Concrete block {GLO}|market for|Cut-off, U
Cover plaster (kg/m^2^)	Cover plaster, mineral {GLO}|market for|Cut-off, U
Adhesive mortar (kg/m^2^)	Adhesive mortar {GLO}|market for|Cut-off, U
Natural Stone (kg/m^2^)	Natural stone plate, polished {GLO}|market for|Cut-off, U
Gravel–crushed (kg/m^2^)	Gravel, crushed {RoW}|market for gravel, crushed|Cut-off, U
PVC (kg/m^2^)	Polyvinylidenchloride, granulate {GLO}|market for|Cut-off, U
Bitumen adhesive (kg/m^2^)	Bitumen adhesive compound, hot {GLO}|market for|Cut-off, U
Aluminium window frame (m^2^/m^2^)	Window frame, aluminium, U = 1.6 W/m2K {GLO}|market for|Cut-off, U
Aluminium door frame (m^2^/m^2^)	Door, outer, wood-aluminium {GLO}|market for|Cut-off, U
Wood door frame (m^2^/m^2^)	Door, inner, wood {GLO}|market for|Cut-off, U
Acrylic filler (kg/m^2^)	Acrylic filler {GLO}|market for|Cut-off, U

**Table 3 ijerph-19-06213-t003:** Materials and sector.

Materials (kg/m^2^)	Sector
Wood	Timber industry
Wood door	
Steel	Steel industry
Aluminium window	
Aluminium door	
Concrete	Chemical and non-metallic inputs
Clay brick	
Flat glass	
Paint	
Tile	
Mortars	
Concrete block	
Cover plaster	
Natural stone plate	
Gravel	

**Table 4 ijerph-19-06213-t004:** The actual Brazilian minimum wage and Gini index.

Year	Minimum Wage (Brazil, USD)	Gini Index
**2002**	40.00	0.589
**2003**	48.00	0.583
**2004**	52.00	0.572
**2005**	60.00	0.570
**2006**	72.00	0.563
**2007**	76.00	0.556
**2008**	83.00	0.546
**2009**	93.00	0.543
**2011**	109.00	0.531
**2012**	124.40	0.530
**2013**	135.60	0.527
**2014**	144.80	0.518
**2015**	157.60	0.491

**Table 5 ijerph-19-06213-t005:** Sector Equivalent Mass (SEM).

	Timber Industry	Steel Industry	Chemical and Non-Metallic Inputs
**Ceramic masonry project**	1.878%	0.812%	97.311%
**Concrete masonry project**	0.283%	1.250%	98.466%

**Table 6 ijerph-19-06213-t006:** Ceramic masonry ceramic masonry and concrete masonry projects’ LCI for the defined functional unit.

Materials	Ceramic Masonry Project	Concrete Masonry Project
Steel (kg/m^2^)	6.37	17.16
Concrete (m^3^/m^2^)	0.28	0.25
Wood (m^3^/m^2^)	0.16	0.01
Clay brick–massive (kg/m^2^)	36.92	13.03
Lime mortar (kg/m^2^)	474.07	281.79
Cement mortar (kg/m^2^)	102.68	83.21
Clay brick–hollow (kg/m^2^)	115.34	0.00
Glass (kg/m^2^)	0.10	0.00
Paint (kg/m^2^)	2.08	2.45
Roof tile (kg/m^2^)	61.92	97.93
Ceramic tile (kg/m^2^)	25.38	21.89
Light mortar (kg/m^2^)	0.00	75.30
Concrete block (kg/m^2^)	0.00	447.03
Cover plaster (kg/m^2^)	0.00	52.23
Adhesive mortar (kg/m^2^)	7.76	6.76
Natural Stone (kg/m^2^)	2.13	0.48
Gravel–crushed (kg/m^2^)	95.42	143.58
PVC (kg/m^2^)	6.05	0.20
Bitumen adhesive (kg/m^2^)	0.23	0.00
Aluminium window frame (m^2^/m^2^)	0.04	0.10
Aluminium door frame (m^2^/m^2^)	0.02	0.08
Wood door frame (m^2^/m^2^)	0.21	0.11
Acrylic filler (kg/m^2^)	1.85	0.00

**Table 7 ijerph-19-06213-t007:** Environmental damages per m^2^ of construction.

Damage Category	Unit	Ceramic Masonry	Concrete Masonry
Damages to human health	HH (DALY)	0.0022	0.0014
Damage to ecosystems	ED (species.yr)	8.09 × 10^−6^	3.11 × 10^−6^
Damage to resources	RA (USD2013)	48.48	32.63

**Table 8 ijerph-19-06213-t008:** Cultural perspective weights (adapted from [61]).

Cultural Perspective	Environmental Dimension	Social Dimension	Economic Dimension
**Hierarchical**	40%	40%	20%
**Egalitarian**	50%	30%	20%
**Individualistic**	25%	55%	20%
**Equal weights**	33.33%	33.33%	33.33%

**Table 9 ijerph-19-06213-t009:** Hierarchical perspective: values of the criteria for determining sustainability.

	Damages to Human Health (Daly)	Damages to Ecosystems (Species.yr)	Damages to Resources (USD 2013)	Weighted Fair Wage Potential	Total Cost (USD/m^2^)
**Ceramic masonry project**	0.0022	8.09 × 10^−6^	48.49	2.608	230.52
**Concrete masonry project**	0.0015	3.11 × 10^−6^	32.63	2.621	221.53

**Table 10 ijerph-19-06213-t010:** Hierarchical perspective: sustainability.

	Damages to Human Health (Daly)	Damages to Ecosystems (Species.yr)	Damages to Resources (USD 2013)	Weighted Fair Wage Potential	Total Cost (USD/m^2^)	Decision Vector
**Criteria vector**	16.00%	16.00%	8.00%	40.00%	20.00%	
**Ceramic masonry project**	40.01%	27.73%	40.23%	49.87%	49.01%	43.81%
**Concrete masonry project**	59.99%	72.27%	59.77%	50.13%	50.99%	56.19%

**Table 11 ijerph-19-06213-t011:** Egalitarian perspective: values of the criteria for determining sustainability.

	Damages to Human Health (Daly)	Damages to Ecosystems (Species.yr)	Damages to Resources (USD 2013)	Weighted Fair Wage Potential	Total Cost (USD/m^2^)
**Ceramic masonry project**	0.036	3.54166 × 10^−5^	49.27	2.608	230.52
**Concrete masonry project**	0.038	2.79975 × 10^−5^	33.31	2.621	221.53

**Table 12 ijerph-19-06213-t012:** Egalitarian perspective: sustainability.

	Damages to Human Health (Daly)	Damages to Ecosystems (Species.yr)	Damages to Resources (USD 2013)	Weighted Fair Wage Potential	Total Cost (USD/m^2^)	Decision Vector
**Criteria vector**	15.00%	25.00%	10.00%	30.00%	20.00%	
**Ceramic masonry project**	50.76%	44.15%	40.34%	49.87%	49.01%	47.45%
**Concrete masonry project**	49.24%	55.85%	59.66%	50.13%	50.99%	52.55%

**Table 13 ijerph-19-06213-t013:** Individualist perspective: values of the criteria for determining sustainability.

	Damages to Human Health (Daly)	Damages to Ecosystems (Species.yr)	Damages to Resources (USD 2013)	Weighted Fair Wage Potential	Total Cost (USD/m^2^)
**Ceramic masonry project**	0.0010	6.21693 × 10^−6^	47.58	2.608	230.52
**Concrete masonry project**	0.0004	1.59988 × 10^−6^	31.72	2.621	221.53

**Table 14 ijerph-19-06213-t014:** Individualist perspective: sustainability.

	Damages to Human Health (Daly)	Damages to Ecosystems (Species.yr)	Damages to Resources (USD 2013)	Weighted Fair Wage Potential	Total Cost (USD/m^2^)	Decision Vector
**Criteria vector**	13.75%	6.25%	5.00%	55.00%	20.00%	
**Ceramic masonry project**	29.54%	20.47%	40.00%	49.87%	49.01%	44.57%
**Concrete masonry project**	70.46%	79.53%	60.00%	50.13%	50.99%	55.43%

**Table 15 ijerph-19-06213-t015:** Equal weights perspective: values of the criteria for determining sustainability.

	Damages to Human Health (Daly)	Damages to Ecosystems (Species.yr)	Damages to Resources (USD 2013)	Weighted Fair Wage Potential	Total Cost (USD/m^2^)
**Ceramic masonry project**	0.0022	8.09 × 10^−6^	48.49	2.608	230.52
**Concrete masonry project**	0.0015	3.11 × 10^−6^	32.63	2.621	221.53

**Table 16 ijerph-19-06213-t016:** Equal weights perspective: sustainability.

	Damages to Human Health (Daly)	Damages to Ecosystems (Species.yr)	Damages to Resources (USD 2013)	Weighted Fair Wage Potential	Total Cost (USD/m^2^)	Decision Vector
**Criteria vector**	11.11%	11.11%	11.11%	33.33%	33.33%	
**Ceramic masonry project**	40.01%	27.73%	40.23%	49.87%	49.01%	44.96%
**Concrete masonry project**	59.99%	72.27%	59.77%	50.13%	50.99%	55.04%

**Table 17 ijerph-19-06213-t017:** Summary table of sustainability assessment.

Perspective	Ceramic Masonry Project	Concrete Masonry Project	More Sustainable Project
Hierarchical	43.81%	56.19%	Concrete masonry project
Egalitarian	47.45%	52.55%	Concrete masonry project
Individualist	44.57%	55.43%	Concrete masonry project
Equal weights	44.96%	55.04%	Concrete masonry project

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
