# Peer review of "Comparison of Brazilian Social Interest Housing Projects Considering Sustainability"

_ijerph, 2022, doi:10.3390/ijerph19106213_

Round 1

Reviewer 1 Report

The manuscript deals with an important topic of showing materials of building construction are related to sustainability evaluation of social interest by comparison of ceramic and concrete masonry projects. Then evaluate sustainability in Brazilian social interest housing projects from LCT. The idea is interesting and worthwhile investigating. The main contribution of this work to this problem is not clearly explained. Furthermore, there are drawbacks in some sections of the manuscript that require a major revision.

Major comment

Improving building energy efficiency is essential for sustainable development; so, this study is needed to assess the energy saving for ceramic and concrete masonry, which were investigated and validated in the literature such as the following.

 https://doi.org/10.1016/j.est.2020.101975

https://doi.org/10.3390/su12093720

Furthermore, the main contribution of this work to this problem is not clearly explained, and for presenting the paper goals, the authors should answer the questions: 1) how is the current research important/relevant? 2) how is it novel and contributes to the state of the art?

Minor comment

  1. Figs. 1-2 are inconsistent form, size and family of the font of their title and legends.
  2. English is ok. However, some typos must be checked.
  3. The figures are ok and they help in the manuscript's description.
  4. The references in the paper should not be older.
  5. What about the limitations of the study? Please, improve the revised version of the manuscript.
  6. The Title could be improved regarding the application of the comparison of ceramic and concrete masonry projects.
  7. The conclusion needs to be well presented for the overall understanding of the research.

Author Response

The manuscript deals with an important topic of showing materials of building construction are related to sustainability evaluation of social interest by comparison of ceramic and concrete masonry projects. Then evaluate sustainability in Brazilian social interest housing projects from LCT. The idea is interesting and worthwhile investigating. The main contribution of this work to this problem is not clearly explained. Furthermore, there are drawbacks in some sections of the manuscript that require a major revision.

Major comment

Improving building energy efficiency is essential for sustainable development; so, this study is needed to assess the energy saving for ceramic and concrete masonry, which were investigated and validated in the literature such as the following.

https://doi.org/10.1016/j.est.2020.101975

https://doi.org/10.3390/su12093720

Furthermore, the main contribution of this work to this problem is not clearly explained, and for presenting the paper goals, the authors should answer the questions: 1) how is the current research important/relevant? 2) how is it novel and contributes to the state of the art?

The authors sincerely appreciate the comments from Reviewer#1 and are very grateful for his/her suggestions and observations. We have revised the original manuscript and added new material in an attempt to reply to his/her concerns and questions cited

The reviewer is totally right when he/she stresses that improving building energy is essential for sustainable development. In this sense, additional text was added to stress this importance, as well as new references, including the above referenced.

Text now sounds (page 21): “As previously stressed, the pre-operational phase was chosen to this study once in this phase the designer can make changes for increasing sustainability. Due to this, the production phase constitutes a promising area of research and contribution to mitigate the impacts. On the other hand, it could be desirable to evaluate materials and systems from a full life cycle view, once a good solution from a production perspective could present a significant impact during the use stage. In addition, it is also important to stress that the inclusion of other important criteria such as the energy efficiency of structural materials can affect the obtained results [56, 62-64].”

 In Conclusions: “In this study, some aspects such as energy efficiency of the materials were not included in the comparisons. On the other hand, due to the pairwise strategy adopted by the multicriteria decision-making method, this and other criteria (including those with more subjectivity) can be easily considered through the proposed methodology.”

Minor comment

  1. Figs. 1-2 are inconsistent form, size and family of the font of their title and legends.

Titles were eliminated from figures and maintained only in the legends.

  1. English is ok. However, some typos must be checked.

An additional revision was made to the text.

  1. The figures are ok and they help in the manuscript's description.

The authors acknowledge the reviewer for his/her opinion.

  1. The references in the paper should not be older.

            References were carefully revised in order to maintain older references only if considered strictly necessary.

  1. What about the limitations of the study? Please, improve the revised version of the manuscript.

            The authors agree with the reviewer that the limitations of the study must be indicated. Some additional text was added to stress these limitations.

Text now sounds (In results and discussion): “In order to enable the proposed study, some limitations and definitions were necessary. Among these, it can be stressed the definition not only of the number of criteria, but also the way they were quantified. For example, the consideration Weighted Fair Wage Potential to Social Life Cycle Assessment was adopted, as in previous studies developed by authors, as proposed in [60]. If other stakeholders are considered (such as the local community) the results can present significant variations [65]”.

  1. The Title could be improved regarding the application of the comparison of ceramic and concrete masonry projects.

            We agree that the title of the paper was very general, and does not evidence that the evaluation was made through a comparison of projects.  In this sense, and considering that the main contribution of the paper, in our opinion, is the inclusion of sustainability in the evaluation of projects, the title was modified to “Comparison of Brazilian Social Interest Housing Projects considering Sustainability”.

  1. The conclusion needs to be well presented for the overall understanding of the research.

            We acknowledge for the review’s observation. The main conclusions are reorganized to facilitate reading and understanding.

Reviewer 2 Report

The objective of this work is interesting for the world of sustainable housing analyzed through its life cycle. However, authors should improve some aspects of it:

Section 1. Introduction: Authors have made a brief presentation of the state of the art, incorporating a sufficient number of scientific papers, but without commenting on the conclusions reached. As a recommendation, they could add more specifically the differences or novel points that their work explores with respect to previous ones. It would also be interesting to include some recent references from the sustainable bricks and artificial aggregates for lightweight concrete sector:
-10.3390/ma13102351
-10.1515/phys-2018-0132
-10.1016/j.suscom.2020.100405
-10.1016/j.jclepro.2015.09.016
-10.3390/app12041917

Section two should be merged with the introduction to follow the authors' guide.

Section 2. Material and Methods: Authors explain, in a logical and accurate way, all the key aspects that every LCA must have, giving the previous explanations of the importance of each one of them and then showing the limits of the system that they have taken into account for the analysis. The choice of calculation methodology is a key aspect to take into account, and the authors have not given reasons for their choice. It would be interesting if they could provide more information in this regard.

Section 3. Results and Discussion: Given the global impact of reading the paper, it would be interesting to add a more globalized currency such as the dollar or the euro. Tables 11, 12, 13 and 16 are misquoted in the text. Table 14 is not mentioned. Revise the text in case there are more inconsistencies.

Section 4. Conclusions: Conclusions are too dense. A division into key points or a clearer separation would be desirable to facilitate reading and understanding.

References: Include some more current reference if any.

In general, I find the paper very complete. The data have been analyzed from different perspectives to verify the accuracy of the results and to be able to provide the most exact information. My congratulations to the authors for their work.

Author Response

The objective of this work is interesting for the world of sustainable housing analyzed through its life cycle. However, authors should improve some aspects of it:

Section 1. Introduction: Authors have made a brief presentation of the state of the art, incorporating a sufficient number of scientific papers, but without commenting on the conclusions reached. As a recommendation, they could add more specifically the differences or novel points that their work explores with respect to previous ones. It would also be interesting to include some recent references from the sustainable bricks and artificial aggregates for lightweight concrete sector:
-10.3390/ma13102351
-10.1515/phys-2018-0132
-10.1016/j.suscom.2020.100405
-10.1016/j.jclepro.2015.09.016
-10.3390/app12041917

The authors sincerely appreciate the comments from the Reviewer. We have revised the original manuscript and added new material in an attempt to reply to his/her concerns and questions cited.

We agree with the reviewer that the paper could add more specifically the differences or novel points with respect to previous ones. In this sense, additional text was added to stress the importance of our study.

Text now sounds (introduction): “This paper proposes a methodology to evaluate sustainability in Brazilian social interest housing projects from Life Cycle Thinking and the Analytic Hierarchy Process. The methodology was applied to two house projects, being: a ceramic masonry project and a concrete masonry project. Based on the results obtained from environmental, economic, and social assessments, the Analytic Hierarchy Process allows the comparison of criteria in order to identify the best global alternative. Based on the obtained results, it is also expected to point out sustainable improvements in housing projects, considering that sustainable projects can make the habitations more appropriate, bringing security, comfort, privacy, and positive impacts to their inhabitants' physical and mental health [37].”

Section two should be merged with the introduction to follow the authors' guide.

Thank you for the suggestion. Section two and the introduction were merged.

Section 2. Material and Methods: Authors explain, in a logical and accurate way, all the key aspects that every LCA must have, giving the previous explanations of the importance of each one of them and then showing the limits of the system that they have taken into account for the analysis. The choice of calculation methodology is a key aspect to take into account, and the authors have not given reasons for their choice. It would be interesting if they could provide more information in this regard.

Authors reorganized the methodology and added additional information in an attempt to detail the study. 

Text now sounds (in Methodology): “The development of the sustainability assessment was initiated with the research and selection of representative projects of social interest of different construction types. Two projects were selected, being a ceramic masonry project and a concrete masonry project. These projects were obtained from public agencies, and their typologies were adopted because they correspond to the vast majority of social interest housing in Brazil. To evaluate the projects regarding LCT, Life Cycle Assessment (LCA), Social Life Cycle Assessment (SLCA), and Life Cycle Cost (LCC) were performed. The following step was the application of the multicriteria decision making method AHP, aiming to identify the best project accordig to several different perspectives. The next items details the adopted methodology and the results obtained from its application.”

Section 3. Results and Discussion: Given the global impact of reading the paper, it would be interesting to add a more globalized currency such as the dollar or the euro. Tables 11, 12, 13 and 16 are misquoted in the text. Table 14 is not mentioned. Revise the text in case there are more inconsistencies.

References to tables were carefully revised, and the currency was changed to the dollar.

Section 4. Conclusions: Conclusions are too dense. A division into key points or a clearer separation would be desirable to facilitate reading and understanding.

We acknowledge the review’s suggestion and divide and reorganize the main conclusions in order to facilitate reading and understanding.

References: Include some more current references if any.

In general, I find the paper very complete. The data have been analyzed from different perspectives to verify the accuracy of the results and to be able to provide the most exact information. My congratulations to the authors for their work.

Thank you again for the important comments and carefully revision

Reviewer 3 Report

The title of the manuscript is very promising but, unfortunately, the reader may be disappointed to read on. There are many threads, but they are incoherent. The analysis of housing projects is necessary, but not from the approach proposed by the authors.

At the very beginning of the manuscript (in the Abstract), the authors state (announce):

“Among the implications of the study carried out here is the advancement of sustainability applied to the construction sector”. Unfortunately, this advancement is not to be seen in this manuscript! This type of analysis is known and has been carried out for many years.

 The authors rightly try to view the problem in many aspects. At least that's what they indicate in the text. Among other things, there is a basic thread, which is about the social aspect:

“… through the inclusion of weighted fair wage potential as a social dimension …”. But it is almost embarrassing that the considerations related to the social dimension (and sustainable development) are reduced only to the statement of the need for fair wages and to the analysis of two wall designs!

In many fragments, the text refers to various reports and actions (noble!), For example:

ReCiPe 2016 Report, (formally it is RIVM Report 2016-0104); SDG 13 - "Climate Action"; The 2030 Agenda (Agenda of the United Nations). These and other terms ad wielded to justify the research. Unfortunately, there is no serious analysis there.

 The list of references quoted at the end of the manuscript should be much richer.

The authors state that their research and their model can be an example for other research and evaluation of solutions in the area of sustainable development and economic matters. Unfortunately, this statement is long overdue. Science and practice have made significant progress in this area, both in terms of sustainable development (also in the construction sector) and the multi-criteria analysis.

I conclude that this manuscript should not be addressed to J. Environ. Res. Public Health, i.e. for the journal dealing with health!

Please note: There are a number of specific comments (regarding the structure, AHP method, data acquisition, criteria, etc.), but I have not listed them here.

Author Response

The title of the manuscript is very promising but, unfortunately, the reader may be disappointed to read on. There are many threads, but they are incoherent. The analysis of housing projects is necessary, but not from the approach proposed by the authors.

At the very beginning of the manuscript (in the Abstract), the authors state (announce):

“Among the implications of the study carried out here is the advancement of sustainability applied to the construction sector”. Unfortunately, this advancement is not to be seen in this manuscript! This type of analysis is known and has been carried out for many years.

The authors sincerely appreciate the comments from the Reviewer. We have revised the original manuscript and added new material in an attempt to reply to his/her concerns and questions cited.

We agree with the reviewer that the title of the manuscript is very broad, and can disappoint the reader. In this sense, the title was changed in an attempt to clarify that the authors are not trying to solve this important and challenging problem but giving a contribution to mitigating the habitational deficit, especially in developing countries. In this sense, to the author’s knowledge, a few studies can be found in literature considering sustainability and decision-making methods.

Based on these considerations, the title of the paper was changed to: Comparison of Brazilian Social Interest Housing Projects considering Sustainability.

 The authors rightly try to view the problem in many aspects. At least that's what they indicate in the text. Among other things, there is a basic thread, which is about the social aspect:

“… through the inclusion of weighted fair wage potential as a social dimension …”. But it is almost embarrassing that the considerations related to the social dimension (and sustainable development) are reduced only to the statement of the need for fair wages and to the analysis of two wall designs!

In many fragments, the text refers to various reports and actions (noble!), For example:

ReCiPe 2016 Report, (formally it is RIVM Report 2016-0104); SDG 13 - "Climate Action"; The 2030 Agenda (Agenda of the United Nations). These and other terms ad wielded to justify the research. Unfortunately, there is no serious analysis there.

As we stressed in the text, the literature review pointed out that most studies consider the environmental dimension as absolute sustainability. For these cases, in the author’s opinion, the study can contribute as a basis for incorporating other dimensions of sustainability, which will generate a more complete sustainability result. To consider the social dimension, several measures can be considered. Authors adopted fair wages mainly due to their importance to the achievement of several Objectives of Sustainable Development (ODS). If other aspects or stakeholders are considered, the presented results can be considerably changed.

Text now sounds (in Results and Discussions): “In order to enable the proposed study, some limitations and definitions were necessary. Among these, it can be stressed the definition not only of the number of criteria, but also the way they were quantified. For example, the consideration Weighted Fair Wage Potential to Social Life Cycle Assessment was adopted, as in previous studies developed by authors, as proposed in [58]. If other stakeholders are considered (such as the local community) the results can present significant variations [65].”

 The list of references quoted at the end of the manuscript should be much richer.

Authors acknowledge the reviewer for his/her suggestion. New references were added to the text in an attempt to enrich the text.

The authors state that their research and their model can be an example for other research and evaluation of solutions in the area of sustainable development and economic matters. Unfortunately, this statement is long overdue. Science and practice have made significant progress in this area, both in terms of sustainable development (also in the construction sector) and multi-criteria analysis.

The authors acknowledge the reviewer’s comment.

I conclude that this manuscript should not be addressed to J. Environ. Res. Public Health, i.e. for the journal dealing with health!

We respectfully disagree with the reviewer about the adequacy of our text to J. Environ. Res. Public Health. If approved, our paper will take part in a special edition called “Edition of Trends in Sustainable Buildings and Infrastructure”. According to the introduction of this special issue, the construction industry is a major contributor to the economic growth of regions through the provision of useful infrastructure and generation of employment, among others, but at the same time is the main source of environmental impacts. In addition, it is important to notice that damage to human health is directly considered in our study.

Please note: There are a number of specific comments (regarding the structure, AHP method, data acquisition, criteria, etc.), but I have not listed them here.

Thank you again for your important contributions and observations. We revised carefully the text in an attempt to improve its quality.

Round 2

Reviewer 1 Report

After revision, the authors have addressed well all my major and minor comments and incorporated the necessary changes in the new revised version of the manuscript, and it is accepted for publication.

Author Response

Once again, the authors appreciate the comments made by Reviewer #1 and believe his/her suggestions and observations have greatly improved the manuscript.

Reviewer 3 Report

====

Author Response

The authors sincerely appreciate the comments made by Reviewer #3 and believe his/her suggestions and observations have greatly improved the manuscript.

This manuscript is a resubmission of an earlier submission. The following is a list of the peer review reports and author responses from that submission.

Round 1

Reviewer 1 Report

In the article the authors state that they use LCT and AHP together. This integration can be traced back to the so-called "mixed method" methodologies, which meet a fair amount of space in the international scientific literature. It should be better clarified in section 3 how the 2 methods are integrated and how they can reciprocally allow evaluations that otherwise, using the methods individually, would not be allowed. From reading your article, it appears that LCT is used to search for parameters to be used for AHP. This aspect is key to your work but does not seem to be clearly evident to me.

The allocation of weights, shown in Table 8, needs to be better justified. The use of the AHP leaves me with some concerns. To determine the preferences of one alternative over another the AHP usually uses the Saaty Scale. However, a more deterministic and less probabilistic approach can be used, and this seems to me to be the route you are taking. You should clarify how you arrive at the percentages in Tables 9, 1-8. I would point out that the numbering of these last 8 tables is incorrect, it should be 10-18. How has AHP been implemented? There are several pieces of software, such as Expert Choice; have you used any of these? There is a lack of robustness analysis on the results and just on the results, a critical analysis on their validity. There must also be a reflection in the conclusions on the usefulness of the procedure you propose in relation to other and more complex evaluations (with more than 2 alternatives and more than 2 criteria).

Reviewer 2 Report

The authors’ decision to focus only on the construction phase of the houses poses a significant problem for the interpretation of the results. The ‘use’ phase of a buildings is generally where the majority of impacts accrue, mostly through energy consumption for heating and cooling as well as maintenance activities. Similarly, impacts accrue during demolition and disposal of materials. The selection of different construction types could have very significant impacts on energy consumption, maintenance and recycling or reuse potential during demolition/disposal. So although one construction type might appear favourable during the construction phase, when considered over the whole life of a building, it might perform far worse than the alternatives. Without this truly whole life approach it is difficult to see how the results of this work can effectively inform design decision-making.  Also, given that the context of this research is social housing, the need to consider how different options effectively deliver affordable comfort over their whole life seems particularly apposite.

The decision to investigate different designs also raises concerns over the results presented. For example, to what degree is the difference in performance simply a consequence of differences in design? To what standard where the houses built in terms of U-values, infiltration rates etc.? Were they both built to the same standard? Also, to what degree do the choices of data from, a wide range of disparate sources, account for the results?

Reviewer 3 Report

There are many studies on LCA-related analysis (and AHP analysis) and this study is one of them. Although combining these two methods have some interest, it is not scientifically interesting as this study evaluated just two examples and the weights for AHP were taken from literature. More original analysis is needed for a peer-reviewed paper.

Reviewer 4 Report

- General comment

In my opinion, the topic of the article is interesting and relevant for future developments.

The article is well structured and the references seem adequate, although there are some that are not referenced in the text and others are missing.

However, I think that extra care should be taken in the presentation of tables so that the document is easier to read and understand, as well as correcting the numbering of some parts that make up the article.

I made some comments to improve the quality of the article. I recommend that authors take into account the observations and resubmit the article.

- Specified comments

Comment 01: page 1, line 34

Where you have "[4,5-8]" you should just put "[4-8]".

Comment 02: page 1, line 35-36

Where you have "[18-21]" won't it be "[17-21]"? Reference [17] is not mentioned in the text.

Please check and correct.

Comment 03: page 2, line 71

You must first indicate the meaning of "TBI" before presenting the acronym, or at least when presenting it for the first time. Like what you did for "LCT".

Comment 04: page 3, line 98

Is it "LLC" or will it be "LCC".

Comment 05: page 5, table 1

You should avoid splitting tables with page change. If this is not possible at least you should avoid splitting the table rows and repeating the header.

This situation appears in several cases.

Comment 06: page 6, line 146

It remains to indicate the figure number. I think it's "Figure 3".

This situation is repeated more often.

Comment 07: page 7 table 2

Same as comment 05.

Comment 08: page 8 table 3

Same as comment 05.

Comment 09: page 9 line 200

The table title/caption must be next to the table and on the same page.

Comment 10: page 15, line 297

It remains to indicate the Table number. I think it's "Table 8".

Please check.

Comment 11: page 16, line 300

There is no reference [61].

Please check and correct.

Comment 12: page 16-19, lines 307, 312, 316, 317, 321, 323, 327, 329, 330, 334, 337, 339, 343, 345, 350, 352 and 357

Table numbering is wrong. For example, where "Table 1" is, it should be "Table 10".

Please check and correct.

Comment 13: page 18, table 6 (table 15)

Same as comment 09.

Comment 14: page 22, line 506

This reference is not indicated in the text.

Comment 15: page 25, line 605

This reference is not indicated in the text.

Round 2

Reviewer 1 Report

Dear Authors,

serious gaps in methodological explanations remain in the text. You declare to use excel, but how?  And ReCiPe 2016? Which is the support of this method? AHP needs robustness analysis. No trace in the text.

Best Regards